# Ameliorative Effects of Flavonoids from *Platycodon grandiflorus* Aerial Parts on Alloxan-Induced Pancreatic Islet Damage in Zebrafish

**DOI:** 10.3390/nu15071798

**Published:** 2023-04-06

**Authors:** Youn Hee Nam, Eun Bin Kim, Ji Eun Kang, Ju Su Kim, Yukyoung Jeon, Sung Woo Shin, Tong Ho Kang, Jong Hwan Kwak

**Affiliations:** 1Department of Oriental Medicine Biotechnology, College of Life Sciences and Graduate School of Biotechnology, Kyung Hee University, Global Campus, Yongin 17104, Republic of Korea; 01030084217@hanmail.net (Y.H.N.); 01073205620@khu.ac.kr (S.W.S.); 2School of Pharmacy, Sungkyunkwan University, Suwon 16419, Republic of Korea; eunbin.kim423@gmail.com (E.B.K.); rkdwlddms789@gmail.com (J.E.K.); cnc0315@daum.net (J.S.K.); jeon1112@skku.edu (Y.J.)

**Keywords:** *Platycodon grandiflorus*, aerial part, flavonoid, pancreatic islet, antidiabetes, K_ATP_ channels, zebrafish

## Abstract

*Platycodon grandiflorus* (balloon flower), used as a food reserve as well as in traditional herbal medicine, is known for its multiple beneficial effects. In particular, this plant is widely used as a vegetable in Republic of Korea. We examined the ameliorative effects of *P. grandiflorus* on alloxan-induced pancreatic islet damage in zebrafish. The aerial part treatment led to a significant recovery in pancreatic islet size and glucose uptake. The efficacy of the aerial part was more potent than that of the root. Eight flavonoids (**1**–**8**) were isolated from the aerial part. Structures of two new flavone glycosides, designated dorajiside I (**1**) and II (**2**), were elucidated to be luteolin 7-*O*-α-L-rhamno-pyranosyl (1 → 2)-(6-*O*-acetyl)-β-D-glucopyranoside and apigenin 7-*O*-α-L-rhamnopyranosyl (1 → 2)-(6-*O*-acetyl)-β-D-glucopyranoside, respectively, by spectroscopic analysis. Compounds **1**, **3**, **4** and **6**–**8** yielded the recovery of injured pancreatic islets in zebrafish. Among them, compound **7** blocked K_ATP_ channels in pancreatic β-cells. Furthermore, compounds **3**, **4**, **6** and **7** showed significant changes with respect to the mRNA expression of *GCK*, *GCKR*, *GLIS3* and *CDKN2B* compared to alloxan-induced zebrafish. In conclusion, the aerial part of *P. grandiflorus* and its constituents conferred a regenerative effect on injured pancreatic islets.

## 1. Introduction

*Platycodon grandiflorus* A. DC., a perennial plant belonging to the Campanulaceae family, is widely distributed and/or cultivated throughout Republic of Korea, Japan, China and Russia [1,2]. The root of *P. grandiflorus* (PgR), a well-known traditional herbal medicine (Platycodi Radix) and food reserve, has been used to treat coughs, sore throats, bronchitis and purulent diseases. It has been used more often for foodstuffs including vegetables than for medicinal purposes in Republic of Korea [2,3]. Previously published phytochemical investigations of *P. grandiflorus* revealed the presence of various triterpenoidal saponins, together with flavonoids, phenolic acids, polyacetylenes and sterols [3,4]. Platycodi Radix and its constituents are known to have various pharmacological effects including anti-inflammatory, immunostimulatory, apophlegmatic, antitussive, antioxidant, antitumor, antidiabetic, antiobesity, hepatoprotective and cardiovascular system support activities [3,4].

Diabetes mellitus is characterized by hyperglycemia following defects in insulin secretion or insulin action. Insulin is synthesized in pancreatic islets (PIs) under the regulation of K_ATP_ channels and voltage-gated Ca^2+^ channels in pancreatic β-cells. When K_ATP_ channels are shut, insulin is secreted, and when they are open, insulin is inhibited [5]. The opening and shutting of K_ATP_ channels involve the cellular utilization of glucose by various tissues, and K_ATP_ channels help regulate blood glucose levels [6]. Glimepiride (GLM), a sul-fonylurea, closes the K_ATP_ channels of pancreatic β-cells and permits Ca^2+^ inflow, inducing insulin secretion [7]. Furthermore, in pre-diabetic and diabetic states, it is known that diazoxide inhibits insulin secretion by opening K_ATP_ channels [8]. Regarding insulin secretion and pancreatic development, genome-wide association studies (GWAS) revealed genes that are related to diabetes mellitus [9].

The antidiabetic activities of Platycodi Radix, which mainly contains triterpenoidal saponins, have been reported [4]. A Platycodi Radix extract shows a hypoglycemic effect in streptozotocin-induced diabetic mice [10]. However, the antidiabetic activity of extracts from the aerial part of *P. grandiflorus* (PgA) has not been previously described. Accordingly, we investigated the antidiabetic activity of *P. grandiflorus* aerial parts in alloxan-induced diabetic zebrafish. Eight flavonoids including two new flavone glycosides were isolated from the aerial part of *P. grandiflorus* by means of activity-guided fractionation and isolation. Their structures were determined from spectroscopic data, and the antidiabetic efficacy of isolated compounds was examined in the diabetic zebrafish model. We confirmed pancreatic islet recovery following treatment with the extract and solvent fractions from the aerial part with isolated compounds and investigated the effect of the isolated compounds on K_ATP_ channels by diazoxide as a K_ATP_ channel opener. Furthermore, the effect of the isolated compounds on *GCK*, *GCKR*, *GLIS3* and *CDKN2B* mRNA expressions in alloxan-induced zebrafish was confirmed using real-time qPCR.

## 2. Materials and Methods

### 2.1. General Experimental Procedures

The melting point was taken using an Electrothermal 9300 (Electrothermal Engineering LTD). UV spectra were obtained from a Hewlett-Packard HP8453 diode array spectrometer. NMR experiments were carried out with a Bruker AVANCE III 700 spectrometer (Ettlingen, Germany). Chemical shifts (*δ*) are reported in parts per million (ppm), referencing the solvent used. ESIMS data were obtained on a Waters Acquity Ultra Performance LC-MS system LCA 048 (Milford, MA, USA). LRFABMS and HRFABMS spectra were obtained on a JMS-700 M Station Mass Spectrometer (JEOL Ltd., Tokyo, Japan). Column chromatography and MPLC were performed on Sephadex LH-20 (25–100 μm, Sigma-Aldrich, St. Louis, MO, USA) and Biotage Isolera One equipped with Biotage^®^ SNAP ULTRA C18 Cartridges (Uppsala, Sweden), respectively. Thin-layer chromatography was conducted on TLC Silica gel 60 F_254_ and TLC Silica gel 60 RP-18 F_254S_ (Merck, Darmstadt, Germany). Fluorescence microscopy was carried out with an Olympus 1X70 microscope (Japan). Image J (NIH, Bethesda, MD, USA) and Focus Lite (Focus Co., Seoul, Republic of Korea) were used for image analysis.

### 2.2. Plant Material

The aerial parts and roots of *P. grandiflorus* were obtained from a farm (Bongsan-myeon, Yesan-gun, Chungcheongnam-do, Republic of Korea) in October 2014. Voucher specimens (root, SKKU-Ph-14-011; aerial part, SKKU-Ph-14-012) were deposited in the School of Pharmacy, Sungkyunkwan University.

### 2.3. Extraction and Isolation

The roots and aerial parts of *P. grandiflorus* were cut into small pieces and lyophilized at −50 °C for 48 h. Dried roots (1.0 kg) and aerial parts (0.82 kg) were extracted twice with ethanol (EtOH) at room temperature and once with EtOH at 60 °C. Each EtOH extract was concentrated on a vacuum evaporator to obtain roots (PgR-EtOH, 77.7 g) and aerial parts (PgA-EtOH, 56.0 g) extracts. The root EtOH extract was subjected to solvent partitioning to yield dichloromethane (PgR-CH_2_Cl_2_ Fr., 2.55 g), ethyl acetate (PgR-EtOAc Fr., 2.80 g), *n*-butanol (PgR-BuOH Fr., 13.72 g) and water (PgR-H_2_O Fr., 52.23 g) fractions (Figure 1A). The EtOH extract of aerial parts was suspended in H_2_O (800 mL) and then sequentially fractionated with dichloromethane, ethyl acetate and *n*-butanol. Each solvent fraction evaporated under reduced pressure to give dichloromethane (PgA-CH_2_Cl_2_ Fr., 3.36 g), ethyl acetate (PgA-EtOAc Fr., 1.25 g), *n*-butanol (PgA-BuOH Fr., 3.91 g) and water (PgA-H_2_O Fr., 72.16 g) fractions (Figure 1B). Among solvent fractions of the aerial part, BuOH and EtOAc solubles were subjected to activity-guided chromatographic separation. The BuOH fraction was chromatographed on a Sephadex LH-20 column eluting with MeOH to yield subfractions B-1 to B-10. Further chromatography of fraction B-7 by RP-18 MPLC (0.1% formic acid (FA) in MeCN/0.1% FA in H_2_O = 5:95 to 40:60, gradient elution) afforded compounds **1** (5.7 mg) and **3** (45.2 mg). Fraction B-5 was rechromatographed on RP-18 MPLC (0.1% FA in MeCN/0.1% FA in H_2_O = 10:90 to 50:50, gradient elution) to yield compounds **2** (3.8 mg) and **4** (18.9 mg). Compound **6** (93.1 mg) was obtained by the recrystallization of fractions B-8 and B-9. The EtOAc fraction was further fractionated on a Sephadex LH-20 column (MeOH only) to yield six subfractions (E-1 to E-6). Fraction E-5 was rechromatographed over RP-18 MPLC using 0.1% trifluoroacetic acid (TFA) in MeCN–0.1% TFA in H_2_O (5:95 to 25:75, gradient elution) as the eluent to obtain compounds **5** (10.5 mg), **6** (86.6 mg) and **7** (43.2 mg). Fraction E-6 was applied over RP-18 MPLC (0.1% FA in MeCN/0.1% FA in H_2_O = 10:90 to 60:40, gradient elution) to yield compound **8** (212.4 mg).

### 2.4. Chemicals and Reagents

Alloxan, glimepiride (GLM) and sea salts were procured from Sigma Chemical Co. (St Louis, MO, USA), while 2-[N-(7-nitrobenz-2-oxa-1,3-diazol-4-yl)amino]-2-deoxyglucose (2-NBDG) was obtained from Invitrogen (Waltham, MA, USA). Furthermore, Diazoxide (DZ) was obtained from Santa Cruz Biotechnology (Santa Cruz, CA, USA).

### 2.5. Zebrafish Care and Embryo Collection

Wild-type zebrafishes were kept in a zebrafish S type system ((W) 1500 X (D) 400 X (H) 2050 mm) (Genomic Design Bioengineering Co., Seoul, Republic of Korea). Spawning cages were prepared by placing 2 pairs of zebrafish females and males overnight. On the following day, zebrafish embryos were collected during the light period at 3 h post-fertilization (hpf) and incubated in Petri dishes with a 0.03% sea salt solution. Embryos were maintained in an incubator at 28.5 °C in 14 h light: 10 h dark cycles. The animals under study were subjected to standardized zebrafish protocols that were approved by the Animal Care and Use Committee of Kyung Hee University, in adherence with ethical guidelines for animal experimentation.

### 2.6. Evaluation of Pancreatic Islets Recovery and Glucose Uptake

The assessment of alloxan-induced pancreatic islet damage in zebrafish was conducted using a previously established protocol [11]. Five-day post-fertilization (dpf), wild-type zebrafishes were carefully transferred into 24-well plates and exposed to 100 μM alloxan for 15 min, followed by replacement with a 0.03% sea salt solution. Following a 6 h incubation period, alloxan-induced zebrafish larvae were treated with GLM, PgA-EtOH, PgR-EtOH and solvent fractions (PgA-CH_2_Cl_2_, PgA-EtOAc, PgA-BuOH, PgA-H_2_O, PgR-CH_2_Cl_2_, PgR-EtOAc, PgR-BuOH and PgR-H_2_O) or isolated compounds at various concentrations for 12 h. After the completion of treatment, zebrafish larvae were stained with 40 μM 2-NBDG for 30 min and rinsed with a 0.03% sea salt solution for 20 min. Fluorescence microscopy was used to capture the pancreatic islet of zebrafish, and subsequently, the pancreatic islet size and fluorescence intensity were evaluated using Focus Lite (Focus Co., Daejeon, Republic of Korea) and Image J software (version 1.50i, National Institutes of Health, Bethesda, MD, USA)

Glucose uptake was defined as glucose transport across cell membranes into the cytosol. We analyzed glucose uptake according to the following equation: relative 2-NBDG glucose uptake = sample (pancreatic islet size × 2-NBDG fluorescence intensity)/normal (pancreatic islet size × 2-NBDG fluorescence intensity).

### 2.7. Action of Diazoxide (DZ) on the Efficacy of Isolated Compounds in Alloxan-Induced Diabetic Zebrafish

The present study employed wild-type zebrafish larvae at five days post-fertilization (dpf), which were distributed across 24-well plates and organized into 16 groups. The groups were as follows: normal, normal treated with DZ or alloxan, alloxan-induced diabetic zebrafish treated DZ, **1**, **1** + DZ, **3**, **3** + DZ, **4**, **4** + DZ, **6**, **6** + DZ, **7**, **7** + DZ, **8** and **8** + DZ. The following dosages were used: isolated compounds (**1**, **3**, **4** and **6**–**8**) at 10 μM each and DZ at 25 μM. The zebrafish larvae were treated with a 100 μM concentration of alloxan for 15 min and then exposed to a 0.03% sea salt solution for 6 h. Alloxan-induced zebrafish larvae were treated (or co-treated) with each compound for 12 h. After the completion of treatment, pancreatic islet images were stained with 40 μM 2-NBDG for 30 min and captured using fluorescence microscopy as described above.

### 2.8. Evaluation of mRNA Expression Using RT-qPCR

The total RNA was extracted from alloxan-induced zebrafish larvae after treatment with or without compounds **3**, **4**, **6** and **7** using a TRIzol reagent (Invitrogen, Carlsbad, CA, USA) according to the manufacturer’s instructions. The quantified total RNA was synthesized to complementary DNA (cDNA) using the Rever Aid First Strand cDNA Synthesis Kit (Thermo Fisher Scientific Korea Ltd., Seoul, Republic of Korea). SYBR Green Master mix (Applied Biosystems, Thermo Fisher Scientific Korea Ltd., Seoul, Republic of Korea) and specific primer pairs (Table 1) were used for RT-PCR, which was carried for 45 cycles of 95 °C for 15 s, 60 °C for 15 s and 72 °C for 20 s. The data were analyzed according to the equation of the −2ΔΔCt method [12], and Beta-actin (β-actin) was used as the endogenous control.

### 2.9. Statistical Analyses

The statistical analysis was executed using GraphPad Prism (version 5). The data were presented as the mean ± standard error of the mean (SEM). To determine statistical significance, repeated one-way analysis of variance (ANOVA) followed by Tukey’s test was conducted. A significance level of *p* < 0.05 was considered statistically significant.

## 3. Results

### 3.1. Efficacy of PgA-EtOH, PgR-EtOH and Their Solvent Fractions in Alloxan-Induced Diabetic Zebrafish

In this study, we aimed to induce pancreatic islet damage in zebrafish via the administration of alloxan, a well-established diabetogenic chemical that has been shown to decrease β-cell mass in pancreatic islets [13]. In a previous investigation, we reported the use of alloxan-induced zebrafish as a type 1 diabetes model with decreased pancreatic islet and β-cell size [11]. To visualize glucose uptake by pancreatic islets, we employed Image J software to generate a histogram that represented pixel intensity (green color) levels ranging from 0 to 255. The average size of pancreatic islets in 5-day-old zebrafish was approximately 1109.89 ± 200.13 μm^2^. However, in zebrafish with alloxan-induced pancreatic islet damage, the size of pancreatic islets was significantly reduced by 33.2% (*p* = 0.0001) compared to the control group. When compared to the alloxan-induced group, the pancreatic islet size in both the GLM-treated and PgR-EtOH extract-treated groups significantly increased by 29.4% (*p* = 0.0019) and 18.0% (*p* = 0.0147), respectively. The islet size also increased significantly in the PgR-CH_2_Cl_2_ and PgR-EtOAc fraction-treated groups (26.7%, *p* = 0.0066, and 28.4%, *p* = 0.0053, respectively). The PgR-BuOH and PgR-H_2_O fraction-treated groups showed no pancreatic islet recovery after alloxan damage. The PgA-EtOH extract-treated pancreatic islet size significantly increased by 25.5% (*p* = 0.0093). The PgA-EtOH treatment led to a 6.0% greater increase in pancreatic islet size than the PgR-EtOH treatment. Furthermore, the PgA-CH_2_Cl_2_, PgA-EtOAc and PgA-BuOH fraction-treated groups showed significantly increased pancreatic islet size (27.6%, *p* = 0.0063; 22.4%, *p* = 0.0491; and 29.2%, *p* = 0.0056, respectively). The pancreatic islet size of the PgA-H_2_O fraction-treated group increased by 4.5% compared to the alloxan group (Figure 2A,C).

Glucose uptake was evaluated in zebrafish treated with PgR-EtOH and PgA-EtOH extracts and associated solvent fractions by detecting the uptake of 2-NBDG fluorescence within the pancreatic islets. After alloxan induction, the uptake of 2-NBDG in pancreatic islets significantly decreased (*p* < 0.0001) compared to the normal group. Glucose uptake by the GLM-treated group was significantly higher (*p* = 0.0024) than that of the alloxan-treated group (negative control). The PgR-CH_2_Cl_2_ and PgR-H_2_O fraction-treated groups also demonstrated significantly greater glucose uptake (*p* = 0.0026 and *p* = 0.0176, respectively) compared to the alloxan group. However, the PgR-EtOH extract-treated group and PgR-EtOAc and PgR-BuOH fraction-treated groups showed no significant difference. The glucose uptake of the PgA-EtOH extract-treated group and the PgA-CH_2_Cl_2_ and PgA-BuOH fraction-treated groups was significantly greater than that of the alloxan-treated group (*p* = 0.0012, *p* = 0.0252 and *p* = 0.0014, respectively). The PgA-EtOAc and PgA-H_2_O fraction-treated groups showed no significant difference. PgA-EtOH led to a significant increase in glucose uptake (31.2%) over PgR-EtOH (Figure 2B).

### 3.2. Efficacy of PgA-EtOAc and PgA-BuOH Fractions in Alloxan-Induced Diabetic Zebrafish

To evaluate the dose-dependency of PgA-EtOAc and PgA-BuOH fractions, we treated alloxan-induced zebrafish larvae with the fractions at concentrations of 0.1, 1, 10, 50 and 100 μg/mL for 12 h. The PgA-EtOAc fraction at 1 μg/mL had the greatest increase in pancreatic islet size (54.13%, *p* = 0.0031) and glucose uptake (*p* = 0.0004); the effect gradually decreased at higher concentrations (Figure 3). Treatment with the PgA-BuOH fraction caused dose-dependent increases between 0.1 and 100 μg/mL; at 100 μg/mL, the pancreatic islet size increased by 97.7% (*p* = 0.0008) compared to the alloxan group, and glucose uptake was also significantly higher than that of the negative control group (*p* = 0.0018) (Figure 4).

### 3.3. Isolation and Identification of Compounds **1**–**8**

The EtOH extract of the aerial part of *P. grandiflorus* was successively partitioned between water and organic solvents (CH_2_Cl_2_, EtOAc and *n*-BuOH). The *n*-BuOH and EtOAc solubles, which showed stronger activity than the other fractions, were fractionated by Sephadex LH-20 column chromatography. Selected fractions were rechromatographed by using RP-18 MPLC to yield compounds **1**–**8**. Six known flavonoids (**3**–**8**) were identified as luteolin 7-*O*-β-D-neohesperidoside (lonicerin, **3**) [14], apigenin 7-*O*-β-D-neohesperidoside (rhoifolin, **4**) [15,16,17], luteolin 7-*O*-(6″-*O*-acetyl)-β-D-glucopyranoside (**5**) [18,19], luteolin 7-*O*-β-D-glucopyranoside (**6**) [19,20], apigenin 7-*O*-β-D-glucopyranoside (**7**) [21] and luteolin (**8**) [14,22] by comparing their spectroscopic data with values from the literature (Figure 5).

Dorajiside I (**1**): yellowish amorphous powder; mp 167–168 °C; UV (MeOH) *λ*max (log ε) 255 (4.33), 266sh (3.88), 350 (4.54); ESIMS (positive ion mode) *m*/*z* 637 [M + H]^+^; FABMS (positive ion mode) *m*/*z* 637 [M + H]^+^; HRFABMS (positive ion mode) *m*/*z* 637.1770 [M + H]^+^ (calcd for C_29_H_33_O_16_, 637.1769); ^1^H NMR (700 MHz, CD_3_OD) and ^13^C NMR (176 MHz, CD_3_OD) data are presented in Table 2.

Dorajiside II (**2**): yellowish amorphous powder; mp 183–184 °C; UV (MeOH) λmax (log ε) 267 (4.30), 335 (4.55); ESIMS (positive ion mode) *m*/*z* 621 [M + H]^+^; FABMS (positive ion mode) *m*/*z* 621 [M + H]^+^, 643 [M + Na]^+^; HRFABMS (positive ion mode) *m*/*z* 621.1818 [M + H]^+^ (calcd for C_29_H_33_O_15_, 621.1819); ^1^H NMR (700 MHz, DMSO-*d*_6_) and ^13^C NMR (176 MHz, DMSO-*d*_6_) data are presented in Table 2.

Compound **1** was obtained as a yellowish amorphous powder. The molecular C_29_H_32_O_16_ formula of **1** was deduced with HRFABMS data (Appendix A). The UV spectrum revealed absorption maxima at 255 and 350 nm, suggestive of a flavone structure. The ^1^H NMR spectrum of **1** (Appendix A) showed signals for a 5,7,3′,4′-tetrahydroxyflavone (luteolin) at *δ*_H_ 6.49 (d, *J* = 2.0 Hz), 6.63 (s), 6.74 (d, *J* = 1.8 Hz), 6.93 (d, *J* = 8.1 Hz), 7.42 (brs) and 7.44 (brd, *J* = 8.1 Hz). Additionally, signals at *δ*_H_ 5.20 (d, *J* = 7.7 Hz) and 5.32 (d, *J* = 1.2 Hz) were assigned to anomeric protons for two sugars; signals at *δ*_H_ 1.38 (d, *J* = 6.2 Hz) and 2.09 (s) were matched with rhamnose methyl and acetyl groups, respectively. The ^13^C NMR spectrum of **1** (Appendix A) exhibited signals for a 5,7,3′,4′-tetrahydroxyflavone (*δ*_C_ 96.2, 101.0, 104.4, 107.2, 114.4, 116.9, 120.6, 123.6, 147.3, 151.4, 159.0, 163.2, 164.4, 167.0 and 184.2), two sugars (*δ*_C_ 18.4, 64.9, 70.2, 71.9, 72.3, 72.3, 74.1, 75.6, 79.1, 79.1, 99.8 and 102.7) and an acetyl group (*δ*_C_ 20.9 and 172.9). The ^13^C NMR data for the two sugars and the acetyl group suggested the presence of 6-*O*-acetyl neohesperidose, and the data fitted well with values from the literature [23]. All proton and carbon signals of **1** were interpreted based on 2D-NMR data (Appendix A). The locations and connectivity of the two sugars and the acetyl group were determined from HMBC correlations (Appendix A) and are summarized in Figure 6. The acetate group, glucose and rhamnose units were assigned to C-6 of glucose, C-7 of luteolin and C-2 of glucose, respectively, because of their HMBC correlations: glucose H-6/C=O of Ac; glucose H-1/luteolin C-7; and rhamnose H-1/glucose C-2 and glucose H-2/rhamnose C-1 (Figure 6). In addition, anomeric configurations for sugar units were determined as β-glucosyl and α-rhamnosyl from the coupling constants of anomeric protons (glucose H-1, *J* = 7.7 Hz; rhamnose H-1, *J* = 1.2 Hz). Thus, the structure of **1** was confirmed as luteolin 7-*O*-α-L-rhamnopyranosyl (1 → 2)-(6-*O*-acetyl)-β-D-glucopyranoside, and it was designated as dorajiside I.

Compound **2**, a yellowish amorphous powder, was assigned the C_29_H_32_O_15_ molecular formula on the basis of HRFABMS data (Appendix A). The UV absorption maxima (267 and 335 nm) of **2** supported the existence of the flavone skeleton. The ^1^H and ^13^C NMR data of **2** (Appendix A) were similar to those of **1** except for the H/C signals for the B-ring of the flavone structure. The ^1^H and ^13^C NMR spectra of **2** exhibited signals for a 5,7,4′-trihydroxyflavone (apigenin) at *δ*_H_ 6.30 (d, *J* = 2.1 Hz, H-6), 6.68 (d, *J* = 2.2 Hz, H-8), 6.81 (s, H-3), 6.87 (d, *J* = 8.8 Hz, H-3′, 5′) and 7.87 (d, *J* = 8.8 Hz, H-2′, 6′); and *δ*_C_ 94.5 (C-8), 99.4 (C-6), 103.2 (C-3), 105.5 (C-10), 116.1 (C-3′, 5′), 120.9 (C-1′), 128.6 (C-2′, 6′), 157.0 (C-9), 161.2 (C-4′), 161.6 (C-5), 162.3 (C-7), 164.4 (C-2) and 182.0 (C-4). The NMR data (Appendix A) implied the presence of an acetyl group, and a glucose and a rhamnose units. The configurations for the glucose and rhamnose units were determined as β- and α-anomeric configurations, respectively, from a large coupling constant (*J* = 7.7 Hz) for glucose H-1 at *δ*_H_ 5.20 and a small coupling constant (*J* = 1.5 Hz) for rhamnose H-1 at *δ*_H_ 5.05. The HMBC correlations (H-1″/C-7, H-2″/C-1‴, H-6″/acetyl C=O and H-1‴/C-2″) revealed that **2** had the same connectivity for the disaccharide and acetyl group as that of **1** (Figure 6 and Appendix A). Therefore, the structure of compound **2** was determined to be an apigenin 7-*O*-α-L-rhamnopyranosyl (1 → 2)-(6-*O*-acetyl)-β-D-glucopyranoside and was named dorajiside II.

### 3.4. Effect of Compounds Isolated from the PgA-EtOAc and PgA-BuOH Fractions on Alloxan-Induced Diabetic Zebrfish

To evaluate the efficacy of isolated compounds, we treated alloxan-induced zebrafish larvae with each compound at 0.1 and 1 μM for 12 h. In alloxan-induced diabetic zebrafish, pancreatic islets decreased by 41.3% (*p* = 0.0002) compared to the normal group, and pancreatic islets in GLM-treated zebrafish increased by 59.5% (*p* = 0.0001) compared with the alloxan group. Compound **1** caused significant recovery at 1 μM (67.4%, *p* = 0.0019), while compound **2** at 0.1 μM increased pancreatic islets by 28.0% (*p* = 0.1032), but this did not reach the level of significance. Compound **3** significantly increased islets at all concentrations, with the largest increase at 1 μM (77.9%, *p* = 0.0002). Compound **4** potently increased islets by 87.5% (*p* < 0.0001) at 1 μM. The group treated with **5** was not meaningfully different compared to the alloxan group. The groups treated with compounds **6** and **7** each had significantly increased islets at 1 μM by 72.2% (*p* = 0.0015) and 41.5% (*p* = 0.0235), respectively. Compound **8** showed strong efficacy at 1 μM (42.9%, *p* = 0.0160) (Figure 7).

### 3.5. Effect of Isolated Compounds on K_ATP_ Channels in Alloxan-Induced Diabetic Zebrafish

The involvement of K_ATP_ channels in the regulation of pancreatic β-cell function was investigated by utilizing DZ as a K_ATP_ channel opener. The pancreatic islet size of the DZ-treated normal group was significantly reduced (by 28.8%, *p* = 0.0256) compared to the normal group without DZ, providing evidence for the action of K_ATP_ channels in this process. In contrast, there was no significant difference in the alloxan group after treatment with DZ. Compounds **1**, **3**, **4**, **6** and **8** were found to be unrelated to K_ATP_ channels, as there was no significant difference observed following the DZ treatment. However, the group treated with compound **6** and DZ (25 μM) showed a weak recovery of pancreatic islets compared to compound **6** alone. Co-treatment with compound **7** and 25 μM DZ resulted in a significantly smaller pancreatic islet size (by 35.7%, *p* = 0.0141) compared to the group treated with compound **7** alone (Figure 8).

### 3.6. Effect of Isolated Compounds on mRNA Expression of GCK, GCKR, GLIS3 and CDKN2B in Alloxan-Induced Diabetic Zebrafish

The mRNA expressions of *GCK*, *GCKR*, *GLIS3* and *CDKN2B*, known to be related to diabetes mellitus, were examined by RT-PCR. As shown in Figure 9, alloxan-induced zebrafish larvae (AX) significantly decreased upon *GCK* and *CDKN2B* mRNA expression and increased upon *GCKR* and *GLIS3* mRNA expression compared with the normal group (NOR). In contrast, most, if not all, sample treatments showed the opposite tendency to the mRNA expression in alloxan-induced zebrafish. In particular, compound **3** showed significant changes in *GCK*, *GCKR* and *GLIS3* mRNA expression compared to alloxan-induced zebrafish.

## 4. Discussion

The zebrafish (*Danio rerio*) has been widely recognized as an established model organism in the field of biomedical research, particularly for investigating the underlying pathophysiology of diverse metabolic disorders. Many methods, including genetic mutation, chemical induction and dietary alteration, have been used for inducing metabolic disease models in zebrafish [24,25]. Recently, studies on pancreas development and modulation in zebrafish have been applied as a model for diabetes and indicate that this model is closely associated with diabetes [26,27]. In addition, it was confirmed that the studies using zebrafish are a good animal model for diabetes research from the efficacy of glimepiride, a common drug for the treatment of diabetes, and it is used as a positive control in previous studies [13,28].

In this study, we determined the recovery effects of PgR-EtOH and PgA-EtOH extracts and their solvent fractions on damaged pancreatic islets in a diabetic zebrafish model. Alloxan significantly decreased the size of pancreatic islets in zebrafish. The measurement of cellular glucose absorption is a widely employed method in diabetes studies, with 2-NBDG being the commonly used fluorescent marker. Glucose uptake is a crucial biological process for maintaining glucose homeostasis, but it is reduced in the presence of pancreatic β-cell damage [29]. In this study, we utilized 2-NBDG to monitor glucose uptake, which is a novel fluorescent marker obtained via the modification of glucose with an amino group at the C-2 position [30]. To serve as a positive control, we employed GLM, which is known to promote insulin secretion by closing K_ATP_ channels [31,32]. The results of our study demonstrated that both pancreatic islet size and glucose uptake were significantly higher in the GLM-treated group when compared to the alloxan group. There have been many reports on the chemical constituents and pharmacological activities of *P. grandiflorus* root, but the aerial part of *P. grandiflorus* has received much less attention [4,33,34]. Accordingly, we sought to investigate the antidiabetic activity of the aerial part. The extract of the aerial part led to a greater increase in pancreatic islet size and glucose uptake than those of the root portion. The EtOAc and BuOH fractions from the PgA-EtOH extract had particularly significant regenerative effects. We found that the dosages for maximum efficacy were 1 μg/mL for PgA-EtOAc and 100 μg/mL for PgA-BuOH.

Two new flavone glycosides, designated as dorajisides I (**1**) and II (**2**), together with six known flavonoids (**3**–**8**) were isolated from the aerial part of *P. grandiflorus* by means of activity-guided fractionation and isolation. The new compounds were identified as luteolin 7-*O*-α-L-rhamnopyranosyl (1 → 2)-(6-*O*-acetyl)-β-D-glucopyranoside (**1**) and apigenin 7-*O*-α-L-rhamnopyranosyl (1 → 2)-(6-*O*-acetyl)-β-D-glucopyranoside (**2**) by spectroscopic analyses. We further investigated the effect of compounds isolated from the EtOAc and BuOH fractions on pancreatic islets in diabetic zebrafish. We focused on compounds **1**, **3**, **4** and **6**–**8** because there was evidence that they caused the recovery of damaged pancreatic islets. The pancreatic islet is primarily composed of beta cells, which account for 70–80% of its total cellular composition [35]. In diabetic patients, the beta cell function weakens, and the mass of islet beta cells decreases [36,37]. Therefore, understanding the regulation of pancreatic β-cell expansion and the role of insulin in glucose homeostasis is crucial. Pancreatic β-cell mass or function may gradually improve when glucose homeostasis and/or normal K_ATP_ channel activity is restored [38,39,40]. We assessed the impact of the compounds on insulin secretion by controlling K_ATP_ channels. To evaluate the effects of **1**, **3**, **4** and **6**–**8** on K_ATP_ channels, we performed co-treatment with DZ following diabetes induction [41,42]. DZ opens K_ATP_ channels in pancreatic β-cells, which inhibits the secretion of insulin in diabetic mice induced by alloxan [43]. Furthermore, a previous study demonstrated that adding coffee, trigonelline or chlorogenic acid along with DZ reduced their beneficial effects in diabetic zebrafish [11]. We observed that co-treatment with DZ decreased the ameliorative effect of **7**. These results suggested that compound **7** acts on K_ATP_ channels and then regulates glucose uptake via the stimulation of insulin secretion. Additionally, the expression of some genes reported to be related to diabetes mellitus by genome-wide association studies (GWAS) [9] was evaluated using RT-PCR in zebrafish larvae. Glucokinase (*GCK*) and glucokinase regulator (*GCKR*) were key factors that controlled the glucose metabolism in pancreatic β cells for maintaining blood glucose homeostasis [44]. In addition, GLIS family zinc finger 3 (*GLIS3*) and cyclin-dependent kinase inhibitor 2B (*CDKN2B*) were associated with pancreatic development for insulin secretion [45,46]. In this study, the alloxan-induced zebrafish (AX) showed significant decreases in the mRNA expression of *GCK* and *CDKN2B* and increases in the mRNA expression of *GCKR* and *GLIS3* compared with the normal group (NOR). In contrast, most, if not all, treatments of compounds showed the opposite tendency relative to the mRNA expression in alloxan-induced zebrafish.

In conclusion, the EtOH extract of the *P. grandiflorus* aerial part and its EtOAc and *n*-BuOH fractions significantly increased the size of pancreatic islets in alloxan-induced diabetic zebrafish. Two new and six known flavonoids (**1**–**8**) were isolated from active fractions. Among the isolated compounds, **1**, **3**, **4** and **6**–**8** possessed significant ameliorative effects in alloxan-induced diabetic zebrafish. The aerial part of *P. grandiflorus* contained flavonoids as antidiabetic components instead of triterpenoid saponins for the root. Furthermore, the active compounds (**1**, **3**, **4** and **6**–**8**) were co-treated with DZ, indicating a relationship with K_ATP_ channels. Compound **7** revealed a regenerative effect on injured pancreatic islets and had blocking K_ATP_ channels. Lastly, compound **7** may act primarily by stimulating insulin secretion, explaining the beneficial effects of the aerial portion of *P. grandiflorus*. In addition, isolated compounds **3**, **4**, **6** and **7** had an effect that significantly changed the mRNA expressions of *GCK*, *GCKR*, *GLIS3* and *CDKN2B*, which are related to diabetes mellitus in the alloxan-induced zebrafish. These results suggest that the *P. grandiflorus* aerial part may have an effect on protecting the pancreatic islet and can improve glucose uptake by blocking the K_ATP_ channel.

## Figures and Tables

**Figure 1 nutrients-15-01798-f001:**
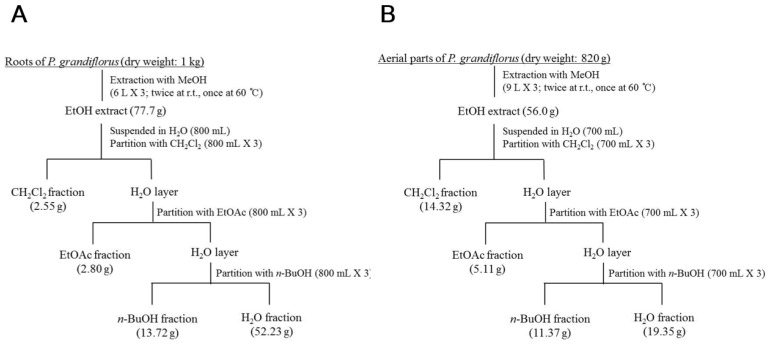
Extraction and solvent partition schemes of the root and aerial part of *P. grandiflorus*. (**A**) Extraction and fractionation scheme of the root; (**B**) extraction and fractionation scheme of the aerial part.

**Figure 2 nutrients-15-01798-f002:**
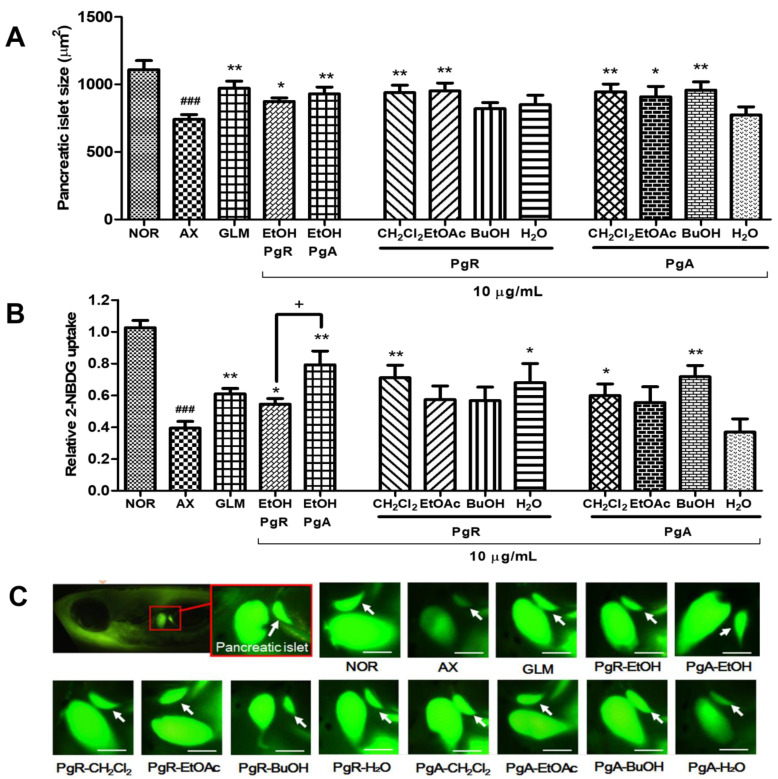
Effect of extracts and solvent fractions from the aerial part and root of *P. grandiflorus* on an alloxan-induced diabetic zebrafish. (**A**) Change in pancreatic islet size caused by GLM, PgR-EtOH and PgA-EtOH extracts (10 μg/mL) and their solvent fractions (10 μg/mL); (**B**) relative 2-NBDG uptake in pancreatic islet caused by GLM, PgR-EtOH and PgA-EtOH extracts and their solvent fractions; (**C**) fluorescent microscopic images of the pancreatic islet (### *p* < 0.001; compared to NOR) (* *p* < 0.05, ** *p* < 0.01; compared to AX) (+ *p* < 0.05). Scale bar = 100 μm.

**Figure 3 nutrients-15-01798-f003:**
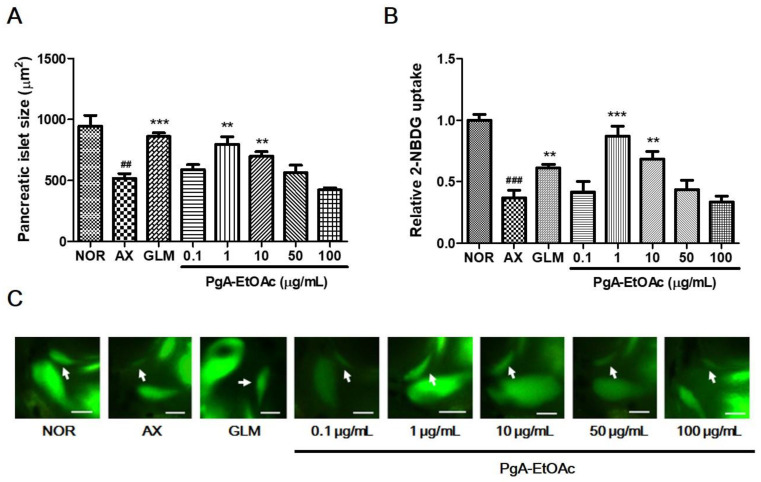
Dose-dependent effects of the PgA-EtOAc fraction on alloxan-induced diabetic zebrafish. (**A**) Change in pancreatic islet size caused by GLM and PgA-EtOAc fraction; (**B**) relative 2-NBDG uptake in pancreatic islet caused by GLM and PgA-EtOAc; (**C**) fluorescent microscopic images of the pancreatic islet. Scale bar = 100 μm (## *p* < 0.01, ### *p* < 0.001; compared to NOR) (** *p* < 0.01, *** *p* < 0.001; compared to AX). Scale bar = μm.

**Figure 4 nutrients-15-01798-f004:**
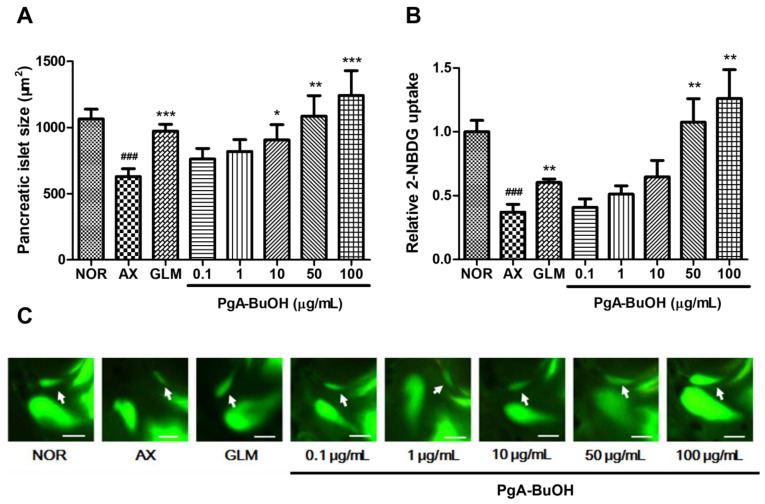
Dose-dependent effects of the PgA-BuOH fraction on alloxan-induced diabetic zebrafish. (**A**) Change in pancreatic islet size caused by the GLM and PgA-BuOH fraction; (**B**) relative 2-NBDG uptake in pancreatic islet caused by GLM and PgA-BuOH; (**C**) fluorescent microscopic images of the pancreatic islet. Scale bar = 100 μm (### *p* < 0.001; compared to NOR) (* *p* < 0.05, ** *p* < 0.01, *** *p* < 0.001; compared to AX). Scale bar = μm.

**Figure 5 nutrients-15-01798-f005:**
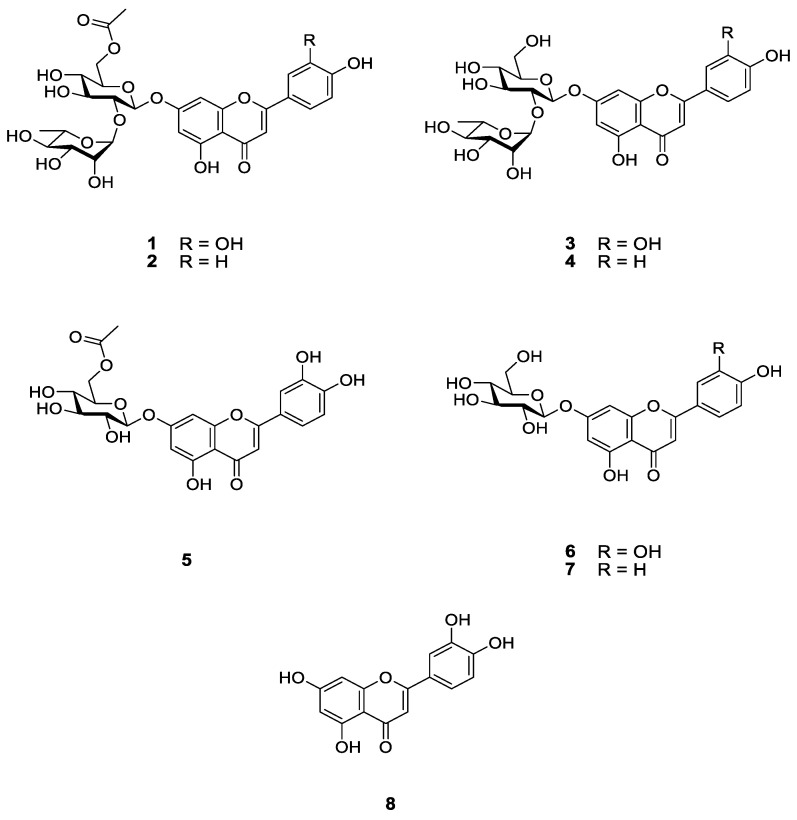
Structures of compounds **1**–**8** from the aerial part of *P. grandiflorus*.

**Figure 6 nutrients-15-01798-f006:**
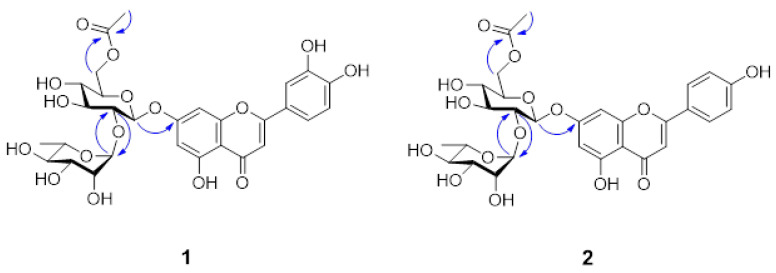
Key HMBC correlations of compounds **1** and **2**.

**Figure 7 nutrients-15-01798-f007:**
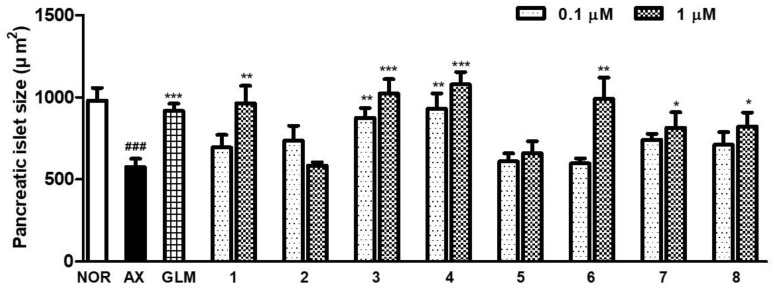
Efficacy of compounds **1**–**8** on alloxan-induced diabetic zebrafish: changes in pancreatic islet size caused by GLM and compounds **1**–**8** (### *p* < 0.001; compared to NOR) (* *p* < 0.05, ** *p* < 0.01, *** *p* < 0.001; compared to AX).

**Figure 8 nutrients-15-01798-f008:**
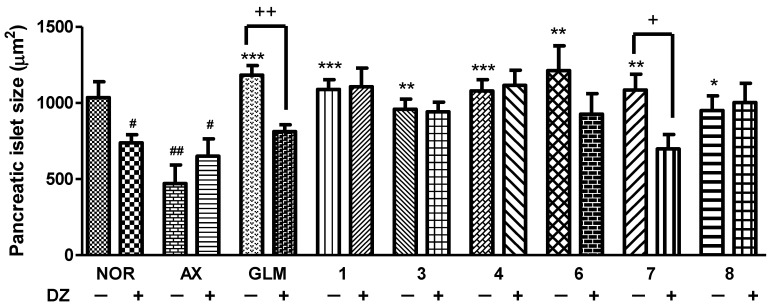
Effect of the isolated compounds for K_ATP_ channel on alloxan-induced diabetic zebrafish: the action of diazoxide (DZ) in the efficacy of compounds **1**, **3**, **4** and **6**–**8** (# *p* < 0.05, ## *p* < 0.01; compared to NOR, (* *p* < 0.05, ** *p* < 0.01, *** *p* < 0.001; compared to AX) (+ *p* < 0.05, ++ *p* < 0.01).

**Figure 9 nutrients-15-01798-f009:**
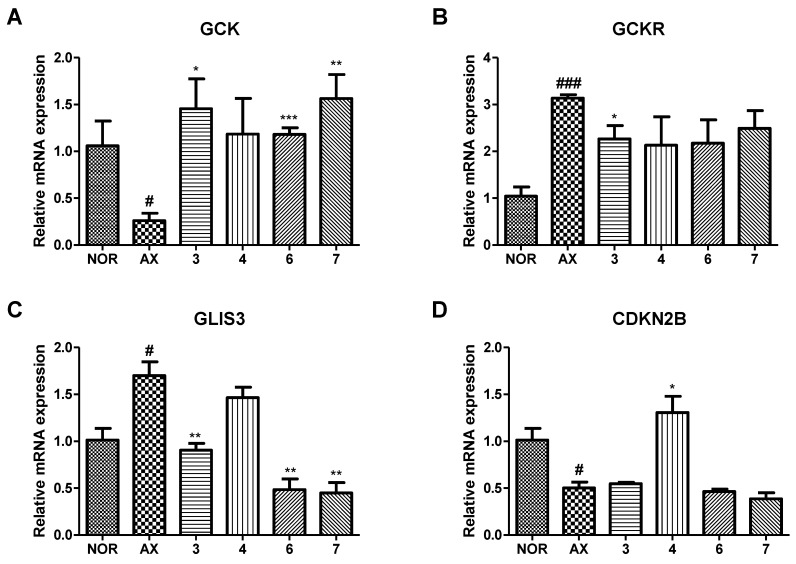
Effect of isolated compounds (**3**, **4**, **6** and **7**) on the mRNA expressions of (**A**) *GCK*, (**B**) *GCKR*, (**C**) *GLIS3* and (**D**) *CDKN2B* in alloxan-induced zebrafish larvae. Values are expressed as means ± SD (# *p* < 0.05, ### *p* < 0.001; compared to NOR) (* *p* < 0.05, ** *p* < 0.01, *** *p* < 0.001; compared to AX); *n* = 3.

**Table 1 nutrients-15-01798-t001:** Primer sequences for RT-PCR.

Gene	Forward Primer	Reverse Primer
*GCK*	5′-ATCCTCATGGTGGACCAA-3′	5′-ATCACCAACCTCGGAGC-3′
*GCKR*	5′-CTGTGAAAGGGCTCTACTGA-3′	5′-AGCAAGAGTACAGCCACACT-3′
*GLIS3*	5′-ATACACTCACACTGCCCTTC-3′	5′-GGACAGTGGATTCTGACAAC-3′
*CDKN2B*	5′-CGGAGTGAATGCCAATCTG-3′	5′-CTGTTCCAGCAGCACAAGAG-3′
*β-actin*	5′-CGAGCTGTCTTCCCATCCA-3′	5′-TCACCAACGTAGCTGTCTTTCT-3′

**Table 2 nutrients-15-01798-t002:** ^1^H and ^13^C NMR data of compounds **1** and **2**.

C/H		Dorajiside I ^1^		Dorajiside II ^2^	
		*δ*_H_ (*J* in Hz)	*δ* _C_	*δ*_H_ (*J* in Hz)	*δ* _C_
	2	-	167.0	-	164.4
	3	6.63s	104.4	6.81 s	103.2
	4	-	184.2	-	182.0
	5	-	163.2	-	161.6
	6	6.49 d (2.0)	101.0	6.30 d (2.1)	99.4
	7	-	164.4	-	162.3
	8	6.74 d (1.8)	96.2	6.68 d (2.2)	94.5
	9	-	159.0	-	157.0
	10	-	107.2	-	105.5
	1′	-	123.6	-	120.9
	2′	7.42 brs	114.4	7.87 d (8.8)	128.6
	3′	-	147.3	6.87 d (8.8)	116.1
	4′	-	151.4		161.2
	5′	6.93 d (8.1)	116.9	6.87 d (8.8)	116.1
	6′	7.44 brd (8.1)	120.6	7.87 d (8.8)	128.6
Glc	1″	5.20 d (7.7)	99.8	5.20 d (7.7)	97.6
	2″	3.73 t (8.2)	79.1	3.47 dd (9.1, 7.8)	76.2
	3″	3.67 t (8.8)	79.1	3.72 m	76.9
	4″	3.40 t (9.4)	71.9	3.13 m	69.9
	5″	3.79 brt (8.4)	75.6	3.43 brt (8.5)	73.8
	6″	4.23 dd (11.9, 7.4)	64.9	4.28 dd (11.9, 1.9)	63.3
		4.48 brd (11.2)		3.99 dd (12.0, 7.4)	
Ac	C=O		172.9		170.2
	CH_3_	2.09 s	20.9	1.94 s	20.6
Rha	1‴	5.32 d (1.2)	102.7	5.05 d (1.5)	100.6
	2‴	3.97 m	72.3	3.63 brs	70.4
	3‴	3.64 dd (9.5, 2.3)	72.3	3.27 m	70.5
	4‴	3.45 t (9.6)	74.1	3.15 m	71.9
	5‴	3.99 m	70.2	3.69 m	68.4
	6‴	1.38 d (6.2)	18.4	1.14 d (6.2)	18.1

^1^ Spectra were taken in CD_3_OD; ^2^ spectra were taken in DMSO-*d*_6._

## Data Availability

The data presented in this study are available from the corresponding author upon request.

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
