# Peer review of "Ameliorative Effects of Flavonoids from Platycodon grandiflorus Aerial Parts on Alloxan-Induced Pancreatic Islet Damage in Zebrafish"

_nutrients, 2023, doi:10.3390/nu15071798_

Round 1

Reviewer 1 Report

This paper investigated the ameliorative effects of flavonoids from a plant on alloxan-induced pancreatic islet damage in zebrafish and identified some active compounds, which may provide a new idea in the treatment of diabetes mellitus. But there are still some questions that need to be clarified:

 1.    Authors should supply the dosage of all the treatments in Fig.1.  

2.    Why did the authors choose PgA-EtOAC and PgA-BuOH rather PgA-CH2Cl2 and PgA-BuOH as the next study to evaluate the dose-dependency, since the effect of PgA-EtOAC was not significant according to the results shown in Fig.1B?

3.    Did the authors design the dosage of PgA-BuOH higher than 100 μg/mL for the dose-dependent experiment?

4.    Line 327: As stated in introduction, the opening of the KATP channels is associated with lower insulin secretion, so maybe we can say the KATP channels reflect the functions of pancreatic islets, but how does the KATP channels influence the size of pancreatic islets? It should be stated.

5.    Line 343: When the authors clarify the effect of isolated compounds on mRNA expression of GCK, GCKR, GLIS3, and CDKN2B, they only chose compound 3,4,6,7. Why?

6.    Line 336-338: You merely detected the size of pancreatic islets without glucose uptake, so the conclusion here is not appropriate.

7.    It’s better to give the practical significance of this article in the end. 

Author Response

Revisions and Responses for the reviewer #1’s comments

  1. Authors should supply the dosage of all the treatments in Fig.1.

Response: Thank you for the reviewer’s comment. We added the information for dosage in the Figure 2 and Legend (10 μg/mL extracts and 10 μg/mL solvent fractions).

  1. Why did the authors choose PgA-EtOAC and PgA-BuOH rather PgA-CH2Cl2 and PgA-BuOH as the next study to evaluate the dose-dependency, since the effect of PgA-EtOAC was not significant according to the results shown in Fig.1B?

Response: AgA-BuOH, which has the highest activity, was first tried to separate the active component, and AgA-EtOAc, which showed a lot of similarities with AgA-BuOH as a result of component comparison, was also subjected to component separation, although it was slightly less active. Although the AgA-CH2Cl2 fraction has excellent activity, it is not included in this study because it contains components of a completely different series that show a big difference in polarity from AgA-BuOH, and we plan to conduct additional research in the future.

  1. Did the authors design the dosage of PgA-BuOH higher than 100 μg/mL for the dose-dependent experiment?

Response: Thank you for the comment. Concentrations higher than 100 μg/ml were not performed because the pancreatic islet size recovered as much as the normal group at 100 μg /ml.

  1. Line 327: As stated in introduction, the opening of the KATP channels is associated with lower insulin secretion, so maybe we can say the KATP channels reflect the functions of pancreatic islets, but how does the KATP channels influence the size of pancreatic islets? It should be stated.

Response: Thank you for this comment. we revised as you recommended.

(Line 404-409) The pancreatic islet is primarily composed of beta cells, which account for 70-80% of its total cellular composition [36]. In diabetic patients, beta cell function weakens, and the mass of islet beta cells decreases [37,38]. Therefore, understanding the regulation of pancreatic β-cell expansion and the role of insulin in glucose homeostasis is crucial. Pancreatic β-cell mass or function may gradually improve when glucose homeostasis and/or normal KATP channel activity is restored [39,40,41].

  1. Kim, S. K.; Hebrok, M. Intercellular signals regulating pancreas development and function. Genes & development. 2001, 15(2), 111-127.
  2. Donath, M. Y.; Halban, P. A. Decreased beta-cell mass in diabetes: significance, mechanisms and therapeutic implications. Diabetologia. 2004, 47, 581-589.
  3. Hanley, S. C.; Austin, E.; Assouline-Thomas, B; Kapeluto, J.; Blaichman, J.; Moosavi, M.; Petropavlovskaia, M.; Rosenberg, L. β-Cell mass dynamics and islet cell plasticity in human type 2 diabetes. Endocrinology. 2010, 151(4), 1462-1472.
  4. De La Vega-Monroy, M. L.; Larrieta, E.; German, M. S.; Baez-Saldana, A.; Fernandez-Mejia, C. Effects of biotin supplementa-tion in the diet on insulin secretion, islet gene expression, glucose homeostasis and beta-cell proportion. The Journal of nutri-tional biochemistry. 2013, 24(1), 169-177.
  5. Tixi-Verdugo, W.; Contreras-Ramos, J.; Sicilia-Argumedo, G.; German, M. S.; Fernandez-Mejia, C. Effects of biotin supple-mentation during the first week postweaning increases pancreatic islet area, beta-cell proportion, islets number, and beta-cell proliferation. Journal of medicinal food. 2018, 21(3), 274-281.
  6. Ashcroft, F. M.; Rorsman, P. KATP channels and islet hormone secretion: new insights and controversies. Nature Reviews Endocrinology. 2013, 9(11), 660-669.

  1. Line 343: When the authors clarify the effect of isolated compounds on mRNA expression of GCK, GCKR, GLIS3, and CDKN2B, they only chose compound 3,4,6,7. Why?

Response: We selected compounds 3 and 4, which showed the highest recovery of pancreatic islet size, compound 7, which showed efficacy on the KATP channel, and compound 6, which did not show statistically effective but showed a clear trend. Therefore, experiments on mRNA expression were conducted for these compounds.

  1. Line 336-338: You merely detected the size of pancreatic islets without glucose uptake, so the conclusion here is not appropriate.

Response: Thank you for the reviewer’s comment. Following your comment 6, line 442-444 has been moved to end of this article.

  1. It’s better to give the practical significance of this article in the end.

Response: Thank you for the comment. We added practical significance of this article to the end.

(Line 442-444) These results suggest that P. grandiflorus Aerial Part may have an effect on protecting the pancreatic islet and improve glucose uptake by blocking the KATP channel.

Reviewer 2 Report

In this study, eight flavonoids including two new flavone glycosides were isolated from the aerial part of Platycodon grandiflorus by means of activity-guided fractionation, and their structures were determined from spectroscopic data, and the antidiabetic efficacy of isolated compounds was assessed in the diabetic zebrafish model. The aerial part and its constituents conferred a regenerative effect on injured pancreatic islets and was more potent than that of the root of P. grandiflorus.

The Materials and Methods are well described and very clear for the comprehension and reproducibility of the work, with one Figure and one Table.

The Results section is very complete, presented with one more Table and eight more Figures, and the most important findings are well connected and discussed with references to other studies in the Discussion.

The attached supporting information is also very detailed and useful, presenting sixteen additional Figures with spectrum analyses of the compounds.

Author Response

Responses for the reviewer #2’s comments

In this study, eight flavonoids including two new flavone glycosides were isolated from the aerial part of Platycodon grandiflorus by means of activity-guided fractionation, and their structures were determined from spectroscopic data, and the antidiabetic efficacy of isolated compounds was assessed in the diabetic zebrafish model. The aerial part and its constituents conferred a regenerative effect on injured pancreatic islets and was more potent than that of the root of P. grandiflorus.

The Materials and Methods are well described and very clear for the comprehension and reproducibility of the work, with one Figure and one Table.

The Results section is very complete, presented with one more Table and eight more Figures, and the most important findings are well connected and discussed with references to other studies in the Discussion.

The attached supporting information is also very detailed and useful, presenting sixteen additional Figures with spectrum analyses of the compounds.

Response: Thank you very much for your kind review and evaluation.
